

# Exploring the interplay between body mass index and passive muscle properties in relation to grip strength and jump performance in female university students

Miguel Ángel Pérez[1], Gabriela Urrejola-Contreras[2], Brian Alvarez[1], Camila Steilen[1], Antonieta Latorre[3] and Maximiliano A. Torres-Banduc[4]

[1] Escuela de Ciencias de la Salud, Carrera de Kinesiología, Universidad Viña del Mar, Viña del Mar, Chile
[2] Escuela de Ciencias de la Salud, Unidad de Ciencias Aplicadas, Universidad Viña del Mar, Viña del Mar, Chile
[3] Centro de Nutrición, Viña del Mar, Chile
[4] Facultad de Ciencias de la Salud, Universidad de Las Américas, Viña del Mar, Chile

Corresponding author
Miguel Ángel Pérez,
m.angelper@gmail.com

## ABSTRACT

**Background:** Women typically have a higher body fat content than men. Fat accumulation is associated with muscle weakness and alterations in mechanical properties. This study aims to determine the relationship between BMI and weight status with the mechanical properties of muscle and tendon. It was hypothesized that the stiffness and tone of the forearm muscle and Achilles tendon would be correlated with weight status and BMI.
**Methods:** A cross-sectional study was conducted with 136 female university students. Grip strength was assessed using a dynamometer, body composition was analyzed through bioimpedance, and countermovement jump performance was evaluated with a force platform. Stiffness and tone were measured using the MyotonPro device. ANOVA was used to compare grip strength and countermovement jump performance according to body composition. The Pearson correlation coefficient was used to examine bivariate associations.
**Results:** Relative grip strength decreased with an increase in fat content, while forearm muscle stiffness and tone decreased with rising weight status and BMI. Stiffness of the Achilles tendon increased with an increase in fat content and showed a significant positive correlation with BMI. Multiple regression analysis revealed a weak correlation between BMI, body composition, and stiffness of the forearm muscles.
**Conclusion:** The results of this study support the notion that the stiffness of the forearm muscles and Achilles tendon is correlated with BMI in young adult women. Furthermore, an increase in body fat percentage is linked to a decrease in mechanical properties and poorer muscle function.

## INTRODUCTION

Obesity is defined as the accumulation of excess fat within the body, leading to various health problems, and has reached pandemic proportions worldwide (*Afshin et al., 2017*). In Chile, 74% of the adult population is classified as overweight or obese (*OECD, 2022*). While the cardiovascular and metabolic consequences of obesity have been extensively studied, its impact on muscle function has received less attention (*Usgu, Ramazanoğlu & Yakut, 2021*). Muscular strength in the upper extremity if often measured *via* handgrip strength (HGS), due to its high reliability and validity (*Vaishya, 2024*). Grip strength results from the combined contraction of the flexor and extensor muscles of the wrist and fingers (*Forman, Forman & Holmes, 2021*). Several forearm flexor muscles, including the flexor pollicis longus, flexor digitorum profundus, flexor carpi ulnaris, and flexor digitorum superficialis (*Oatis, 2009*), contribute to wrist flexion and grip force production. HGS is known to be influenced by weight, height, age, and body mass (*Roman-Liu, Tokarski & Mazur-Różycka, 2021*). However, to our knowledge, there is limited information regarding the association between the viscoelastic properties of the forearm muscles, HGS, and body mass index (BMI).

Muscle stiffness is recognized as a marker of muscle performance and joint stability during functional movements. Indeed, the stiffness and tone of the forearm flexor muscles have been positively correlated with handgrip strength (*Çevik Saldıran, Kara & Kutlutürk Yıkılmaz, 2022*). Stiffness depends on muscle structure, including length and cross-sectional area, as well as the forces applied and the intrinsic material properties of the muscle (*Baumgart, 2000*). Furthermore, stiffness may significantly impact force production within muscles (*Bizzini & Mannion, 2003*; *Pożarowszczyk et al., 2018*). While some stiffness may be favorable for performance, either too much or too little stiffness can lead to injury (*Bret et al., 2002*; *Brughelli & Cronin, 2008*; *Maciejewska-Skrendo et al., 2020*; *Miyamoto, Hirata & Kanehisa, 2015*).

Several studies have demonstrated that greater muscle strength and higher tendon stiffness positively influence jumping performance (*Arampatzis et al., 2001*; *Brughelli & Cronin, 2008*; *Butler, Crowell & Davis, 2003*; *Waugh, Korff & Blazevich, 2017*). Most research on the biomechanical properties of muscles suggests that higher stiffness is advantageous for activities involving a fast stretch-shortening cycle and high movement velocity (*Brughelli et al., 2008*), such as jumping actions (*Flanagan, 2007*). Vertical jumping tests are widely used to evaluate both simple and complex tasks, as well assess lower limb muscular strength and power performance (*Kitamura et al., 2017*; *Petrigna et al., 2019*). The two commonly used tests considered reliable and valid for monitoring jump performance and lower limb strength are the squat jump (SJ) and the countermovement jump (CMJ). As mentioned earlier, force generation depends on both internal and external factors. Regarding internal factors, muscle mass and volume positively influence strength, while intramuscular fatty infiltration may negatively impact muscle strength (*Staron et al., 2000*). It has been observed that greater muscle mass is associated with increased stiffness, and muscles with greater passive stiffness tend to have more muscle mass (*Chleboun et al., 1997*). Fat accumulation within skeletal muscle is linked to muscle weakness and poor

function (*Goodpaster et al., 2006*). Therefore, an increase in weight status or BMI is associated with a greater likelihood of low muscle strength in healthy young adult women (*Sung et al., 2022*). Excessive fatty infiltration results in an increase in fibrous components (and a decrease in contractile elements), which, along with the reduction in the size and number of muscle fibers, may lead to changes in the viscoelastic properties of the muscle (*Usgu, Ramazanoğlu & Yakut, 2021*).

In women, strength performance is lower compared to men due to higher body fat content, which accounts for the observed differences between the sexes (*Mansour et al., 2021*). Additionally, the viscoelastic properties of muscles are significantly lower than those in men (*Hoffman et al., 2021*). Therefore, women are a suitable group for investigating the following research question: What is the effect of fat content on the viscoelastic properties of muscle and tendon? Researchers have attempted to correlate stiffness with BMI; however, many studies have either used samples that do not include all BMI categories or have included a wide age range (*Kocur et al., 2017*; *Römer et al., 2023*; *Usgu, Ramazanoğlu & Yakut, 2021*). It has been found that in older adults (50–80 years), stiffness does not correlate with body mass or BMI (*Tomlinson et al., 2021*). Consequently, this will be the first study to correlate the viscoelastic properties of muscle and tendon with BMI and body fat content in a group of healthy young adult women.

Previous studies have indicated that the myotonometer is highly reliable for measuring skeletal muscle viscoelastic parameters in healthy individuals and those with various diseases (*Çevik Saldıran, Kara & Kutlutürk Yıkılmaz, 2022*; *Chuang, Wu & Lin, 2012*; *Fröhlich-Zwahlen et al., 2014*; *Marusiak et al., 2010*). Additionally, muscle tone and stiffness have been found to be positively correlated with lower limb muscle strength and upper limb hand motor function (*Fröhlich-Zwahlen et al., 2014*).

Therefore, the aims of this study were to determine the relationship between BMI and the mechanical properties of muscle and tendon and compare these properties according to weight status. It was hypothesized that (1) greater stiffness and tone of the palmaris longus and flexor digitorum superficialis muscles would be correlated with lower weight status or BMI, and (2) greater stiffness and tone of the Achilles tendon would be correlated with higher weight status or BMI.

## MATERIALS AND METHODS

### Participants

A cross-sectional, quantitative correlational study was conducted among 136 female university students, aged between 18 and 25 years (mean age 19.47 ± 1.5), who were recruited from the University of Viña del Mar. Only women were recruited, as they potentially have a different distribution of adipose tissue in the muscular system compared to men (*Gallagher et al., 1996*). The participants' mean weight was 68.89 ± 14.8 kg, height was 1.60 ± 5.6 m, and BMI was 26.75 ± 5.4 kg/m$^2$, as summarized in Table 1. The healthy young adults were fully informed about the study's purpose, and written informed consent was obtained from each participant. Ethical approval for this study was obtained from the institutional ethics committee of the Universidad Viña del Mar (Reference Number: CEC-UVM 24-23). Participants underwent a range of anthropometric and physical

**Table 1 Demographics, body composition, and absolute and relative to body mass handgrip strength of female subjects.**

| Parameters young adult (18–25) | Normal ($n = 16$) | Normal-obesity ($n = 35$) | Overweight ($n = 60$) | Obesity ($n = 25$) | Mean ± SD |
|---|---|---|---|---|---|
| Age (years) | 18.6 ± 0.9[a] | 19.4 ± 1.5[a] | 19.7 ± 1.7[a] | 19.6 ± 1.5[a] | 19.47 ± 1.5 |
| Body mass (kg) | 52.4 ± 6.4[a] | 58.2 ± 5.9[a] | 69.9 ± 5.9[,b] | 90.1 ± 14.7[c] | 68.89 ± 14.8 |
| Height (m) | 161.0 ± 8[a] | 158.9 ± 5.6[a] | 160.1 ± 5.3[a] | 160.8 ± 5.8[a] | 160.1 ± 5.6 |
| Body mass index (kg/m$^2$) | 19.9 ± 2.0[a] | 22.9 ± 1.3[b] | 27.3 ± 1.4[c] | 35.1 ± 5.6[d] | 26.75 ± 5.4 |
| Body fat mass (kg) | 12.8 ± 2.6[a] | 19.6 ± 2.8[b] | 27.1 ± 3.8[c] | 41.7 ± 10.7[d] | 26.15 ± 10.3 |
| Percent body fat (%) | 24.4 ± 3.4[a] | 33.8 ± 2.7[b] | 38.6 ± 3.4[c] | 45.3 ± 4.7[d] | 36.91 ± 6.8 |
| Fat free mass (FFM) (kg) | 39.5 ± 5.0[a] | 38.4 ± 4.0[a] | 42.8 ± 3.8[b] | 49.4 ± 5.9[c] | 42.49 ± 5.7 |
| Skeletal muscle mass (SMM) (kg) | 21.4 ± 3.1[a] | 20.8 ± 2.4[a] | 23.5 ± 2.3[b] | 27.4 ± 3.6[c] | 23.27 ± 3.5 |
| Handgrip strength (kg) | 22.2 ± 5.4[a,b] | 21.1 ± 5.9[a] | 22.8 ± 4.5[a,b] | 25.1 ± 7.0[b] | 23.09 ± 5.2 |
| Handgrip relative to body mass | 1.1 ± 0.2[a] | 0.9 ± 0.2[b] | 0.8 ± 0.2[b,c] | 0.7 ± 0.2[c] | 0.88 ± 0.2 |
| CMJ Force (N) | 1,079.0 ± 241.2[a] | 1,176 ± 241.7[a] | 1,373 ± 244.3[b] | 1,767 ± 438[c] | 1,362 ± 361.5 |
| CMJ Force relative to body mass (N/kg) | 28.1 ± 8.3[a] | 26.3 ± 9.5[a] | 25.3 ± 8.3[a] | 25.04 ± 9.9[a] | 25.80 ± 8.8 |
| CMJ Power (W) | 291.2 ± 120.1[a,b] | 241.3 ± 86.03[a] | 257.5 ± 79.9[a] | 327.0 ± 89.70[b] | 268.1 ± 94.5 |
| CMJ Power relative to body mass (W/kg) | 5.4 ± 1.9[a] | 4.1 ± 1.46[b] | 3.7 ± 1.2[b] | 3.6 ± 0.9[b] | 4.0 ± 1.4 |

**Note:**
Countermovement jump (CMJ). Significant differences were assessed by one-way ANOVA. The letters a, b, c, and d indicate statistically significant difference at $p < 0.05$ within each row comparison between groups. Parameters with no common letters are significantly different ($p < 0.05$).

performance assessments in a single testing session. First, students who voluntarily participated attended the science school laboratory, where anthropometric parameters were measured. Next, an operator experienced in using the MyotonPro equipment assessed the mechanical properties of the forearm muscles and the Achilles tendon. Then, after explaining the procedure, grip strength in both extremities was measured. Following this, the jump test was conducted, and finally, impedance analysis measurements were performed. Participants were instructed on the correct execution of the CMJ and given the opportunity to practice jumps with feedback from an instructor before performing the actual jumps.

### Myotonometric assessment of muscle mechanical properties

After collecting anthropometric measures, muscle stiffness (the tissue's resistance to force that changes its shape) and oscillation frequency (as an indication of tone, or the resting level of tension in the tissue) (*Aird, Samuel & Stokes, 2012*; *Chuang, Wu & Lin, 2012*) in the forearm flexor muscles (including the palmaris longus and flexor digitorum superficialis) and the Achilles tendon were assessed using the MyotonPRO device (Myoton AS, Tallinn, Estonia). Briefly, the MyotonPRO probe was placed perpendicular to the skin surface overlying the muscle belly being measured. Once the device was held stable in position, it applied an automatic preload of 0.18 N to register the natural damping oscillation of the muscle through the overlying skin and subcutaneous tissue. Additionally, the device exerted an automatic mechanical impulse with a duration of 15 ms to the contact area. Following this, a light, quick-release mechanical force of 0.4 N was applied for 15 ms to induce muscle deformation. Subsequently, the integrated accelerometer recorded the

muscle response as dampened natural oscillation, from which stiffness and tone were computed (*Chuang, Wu & Lin, 2012*). The examination began with the flexor digitorum superficialis (FDS), followed by the palmaris longus muscle, and concluded with an assessment of the Achilles tendon. Measurements were performed on the dominant hand and foot.

## Procedures

The location of the flexor digitorum superficialis (FDS) muscle was determined based on anatomical landmarks. Participants were comfortably seated upright with their arms resting on the table. The elbow was bent at approximately 120 degrees, the forearm was supinated, the hand was positioned palm up, and the wrist and fingers were relaxed. To locate the FDS muscle, we identified the point two-thirds of the distance from the elbow along the line between the medial epicondyle of the humerus and the palmaris longus tendon. Participants were instructed to oppose the thumb and little finger with the wrist in slight flexion to help identify the palmaris longus tendon. Subsequently, we palpated approximately 10 cm from the ulnar styloid process to locate the belly of the FDS muscle while asking them to flex the middle and ring fingers (*Tantipoon et al., 2023*). The measurement site was marked with a dermatological pen as a black circle (Fig. 1). For the examination of the palmaris longus muscle, we targeted the point one-third of the distance from the elbow along the line between the medial epicondyle of the humerus and the palmaris longus tendon, marked by a blue circle (Fig. 1). Finally, this area was marked with a non-permanent marker to designate it as the point for myoton measurements.

## Handgrip strength

For the next step, individual maximum grip force was assessed using a hydraulic dynamometer with an adjustable grip (Baseline® model). Measurements were first performed on the right side, followed by the left side of the body. Participants were instructed to stand upright with their arms at their sides and to squeeze with maximum force for 3 to 5 s, guided by standardized verbal cues from the researcher. The value obtained before resetting the dynamometer to 0 was recorded. This process was repeated three times for each hand, with a 1-min rest between each trial. The highest score from the three trials of the dominant hand was used in the analysis. The right hand was dominant in 98.5% of participants.

## Anthropometric measures

For body measurements, standardized techniques were employed. Two measurements of each parameter were taken, and the mean of these measurements was calculated. Weight was determined using a SECA scale (model 700, precision 50 g), with the subject in light clothing centered on the scale's plate. Height was measured using a SECA stadiometer while the subject stood with shoes removed, shoulders relaxed, and facing away from the wall. BMI was calculated from weight (kg) and height (m) squared and expressed in $kg/m^2$. BMI categories followed international adult standards: underweight (BMI < 18.5), normal weight (BMI = 18.5 to 24.9), overweight (BMI = 25.0 to 29.9), and obesity

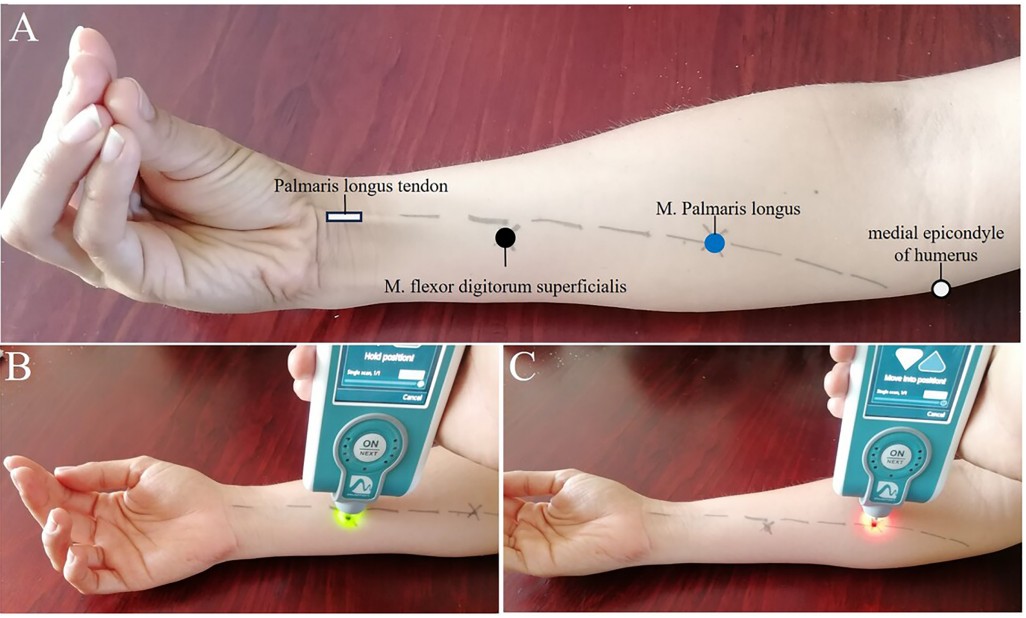

**Figure 1 Experimental setup.** (A) Setup for MyotonPRO measurement. (B) Measurement point on the flexor digitorum superficialis muscle (black circle). (C) Palmaris longus muscle landmark determination (blue circle).

(BMI > 30 kg/m²). Normal-weight obesity refers to individuals with normal body weight and BMI but a high body fat percentage (≥30% body fat; BMI: 18.5–24.9 kg/m²), as defined in a previous study (*Kapoor et al., 2019*). Body composition was assessed using the InBody 270 direct segmental multi-frequency bioelectrical impedance analysis device (InBody Co. Ltd, South Korea). Measurements were conducted in temperature-controlled labs. Body fat mass and lean mass were measured using a standardized protocol with an eight-electrode multi-frequency segmental system, which underwent regular servicing and calibration.

## Jump performance

A force platform (Art Oficio, PF-4000/50; Chile) was used to measure the countermovement jump (CMJ). Participants underwent a warm-up prior to the test. During the test, each participant assumed a standing position with feet parallel and shoulder-width apart, knees extended, and arms at their sides. After a quick downward movement, they flexed their knees and hips, followed by a rapid extension of the knees to achieve a maximum vertical jump. Participants performed three CMJ repetitions with a 1-min rest between each, and the best performance was used for subsequent statistical analyses.

## Statistical analysis

The Shapiro-Wilk test was used to assess the normality of the distribution, while Bartlett's test was employed to analyze the homogeneity of variances. If a variable did not follow a normal distribution or if variances were not homogeneous, a log transformation of the data was performed, followed by a back transformation to present the results. Continuous data were expressed as means and standard deviations. Differences in handgrip strength (HGS),

stiffness, and tone of forearm muscles and Achilles tendons according to weight status (normal, normal-obesity, overweight, obesity) were analyzed using a one-way analysis of variance (ANOVA) followed by Tukey's *post-hoc* tests. Potential differences in countermovement jump (CMJ) parameters according to weight status were also analyzed using ANOVA. The relationships between numerical variables were evaluated using Spearman's correlation. A Spearman's correlation coefficient of 0.00–0.10 was interpreted as indicating very weak or no correlation, 0.10–0.39 as weak correlation, 0.40–0.69 as moderate correlation, 0.70–0.89 as high correlation, and 0.90–1.00 as very strong correlation (*Schober, Boer & Schwarte, 2018*). Multiple regression analysis was conducted to examine the effects of BMI and body composition on the stiffness and tone of the forearm muscles and Achilles tendons. In this analysis, BMI and body composition were considered independent variables, while the mechanical properties of the muscle or tendon were treated as dependent variables. A general linear model analysis was used, with the alpha level set at $p < 0.05$, and all hypothesis tests were two-tailed. Statistical analysis was performed using GraphPad version 8.01 for Windows.

## RESULTS

Table 1 presents the descriptive statistics of the female university sample, categorized into four groups based on weight status. Age and height exhibit a homogeneous distribution across these categories. As expected, weight, body mass index, body fat mass, percent body fat, fat-free mass, and skeletal muscle mass all show significant increases from the healthy weight group to the obese group. Grip strength is significantly higher in obese participants compared to those with normal-obesity weight. However, when expressed relative to body mass, grip strength shows a significant decrease from the normal weight group to the obese group. For CMJ force and power, there is a significant increase from the normal weight group to the obese group. However, when expressed as relative values, CMJ force and power show a significant decrease from the normal weight group to the obese group (Table 1).

### Handgrip strength, weight status, body mass index, and body composition

Relative HGS, calculated and analyzed by weight status, showed lower values in higher adiposity categories (Fig. 2A). Significant differences were observed between the normal weight and normal-obesity groups (normal weight: $1.11 \pm 0.2$ kg/kg/m$^2$; normal-obesity: $0.93 \pm 0.2$ kg/kg/m$^2$, $p < 0.01$) ($F_{(132,2)} = 13.47$, $p < 0.0001$), as well as between the normal weight group and both the overweight and obesity groups (normal weight: $1.11 \pm 0.2$ kg/kg/m$^2$; overweight: $0.84 \pm 0.2$ kg/kg/m$^2$; obesity: $0.73 \pm 0.2$ kg/kg/m$^2$, $p < 0.0001$). A significant moderate negative correlation was found between relative handgrip strength and BMI, which is closely related to fat tissue ($r = -0.44$, $p = 0.0001$) (Fig. 2B). A moderate negative correlation was also observed between BMI and body fat mass ($r = -0.51$, $p = 0.0001$) (Fig. 2C). Additionally, there was a weak negative correlation between relative handgrip strength and fat-free mass ($r = -0.29$) (Fig. 2D).

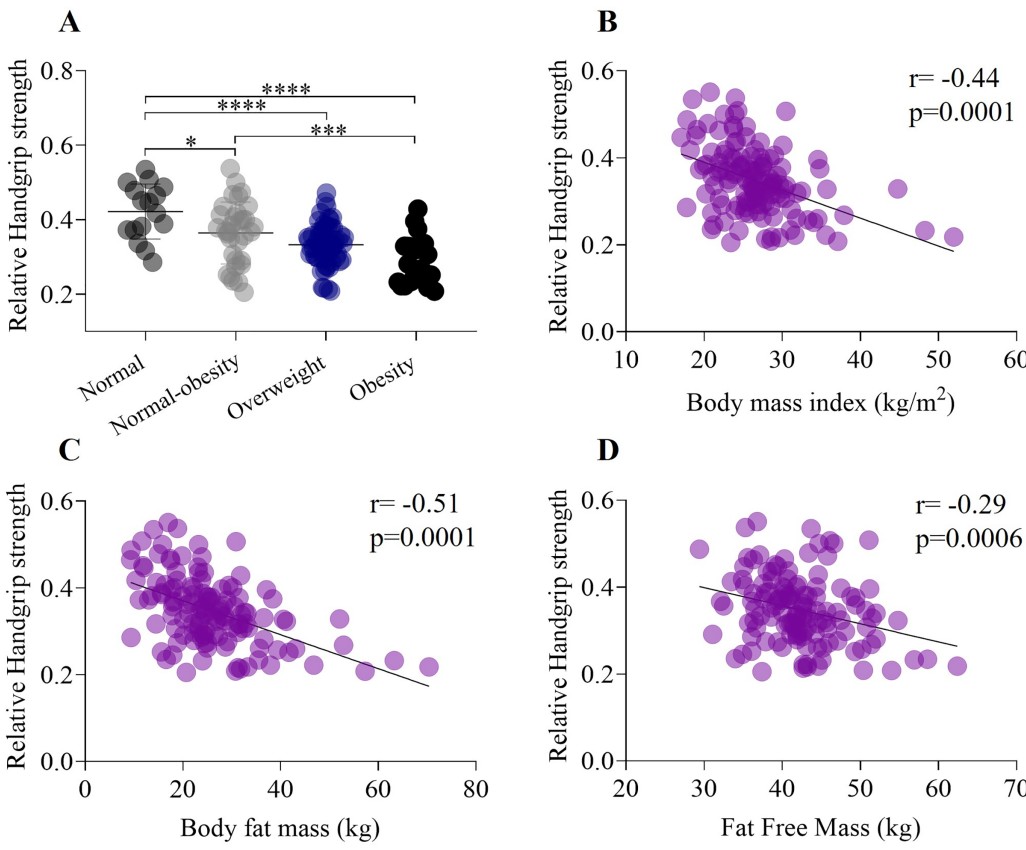

**Figure 2 Association between handgrip strength, weight status, body mass index, and body composition.** (A) Relative grip strength exhibits a significant decrease from the healthy weight status to obesity category. In addition, there was a significant differences between normal weight and normal-obesity category (normal weight: 0.42 ± 0.07; normal-obesity: 0.29 ± 0.06, $p < 0.0001$). (B) Relative Handgrip strength have a negative moderate correlation (r = −0.44) with body mass index. (C) Relative handgrip strength have a negative moderate correlation (r = −0.51) with body fat mass. (D) Relative Handgrip strength have a negative weak correlation (r = −0.29) with fat free mass. *$p < 0.5$, ***$p < 0.001$, ****$p < 0.0001$.

## Mechanical properties, weight status and body mass index

A one-way ANOVA showed a significant decrease in the stiffness of the palmaris longus muscle from the normal weight group to the obesity group (normal weight: 265.7 ± 35.4 N/m; obesity: 233.3 ± 33.3 N/m, $p < 0.01$) ($F_{(132,2)} = 6.912$, $p < 0.0001$) (Fig. 3A). There was a significant weak negative correlation between the stiffness of the palmaris longus muscle (r = −0.35, $p = 0.0001$) and BMI (Fig. 3B). Similarly, one-way ANOVA showed a significant decrease in the tone of the palmaris longus muscle from the normal weight group to the obesity group (normal weight: 15.3 ± 1.3 Hz; obesity: 14.32 ± 1.1 Hz, $p < 0.5$) ($F_{(132,2)} = 4.63$, $p < 0.001$) (Fig. 3C), and the tone of the palmaris longus showed a negative weak correlation (r = −0.31) with BMI (Fig. 3D). A one-way ANOVA revealed a significant decrease in the stiffness of the flexor digitorum superficialis from the normal weight group to the obesity group (normal weight: 298.4 ± 39.8 N/m; obesity: 273.0 ± 35.0 N/m, $p < 0.5$) ($F_{(132,2)} = 4.5$, $p < 0.01$) (Fig. 3E). There was a significant weak negative

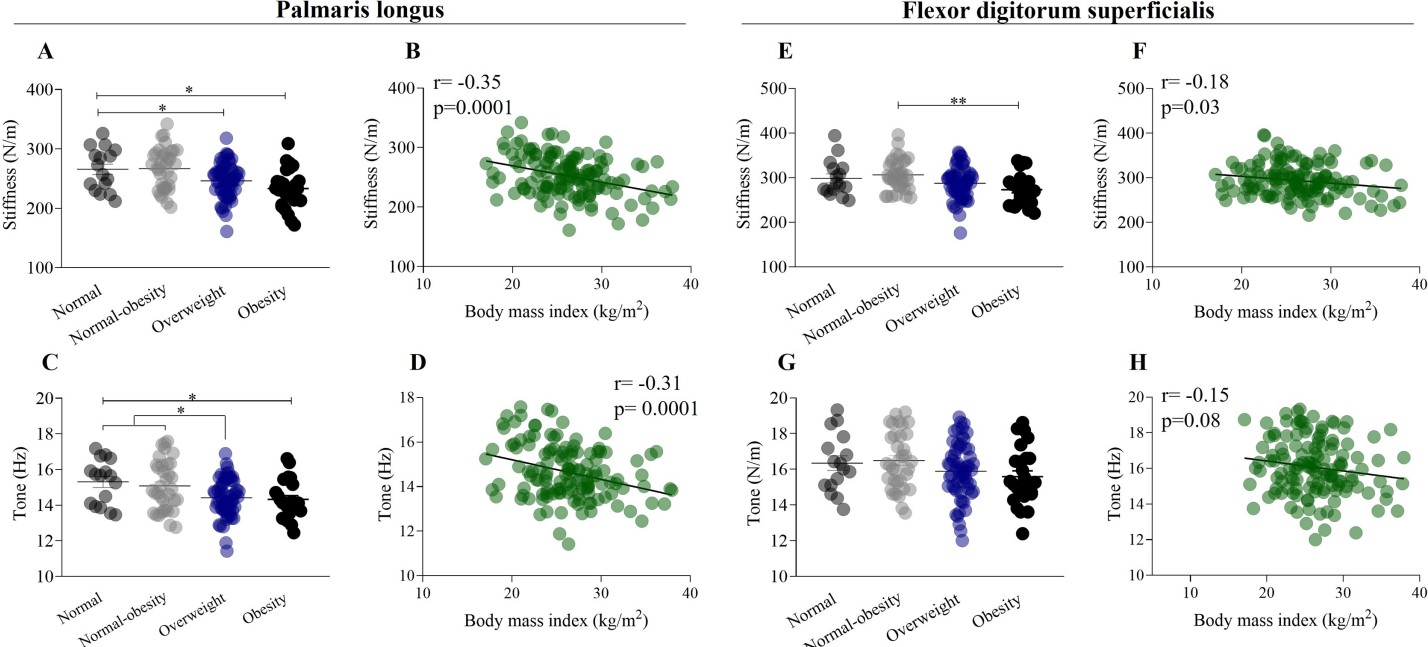

**Figure 3 Mechanical properties of forearm flexor muscles in relation to weight status and body mass index.** (A) Stiffness of palmaris longus exhibit a significant decrease from normal weight to obesity category (normal weight: 265.7 ± 35.4 N/m; obesity: 233.3 ± 33.3 N/m). (B) Stiffness of palmaris longus have a negative weak correlation (r = −0.35) with body mass index. (C) Tone of the palmaris longus muscle exhibit a significant decrease from normal weight to obesity category (normal weight: 15.3 ± 1.3 Hz; obesity: 14.32 ± 1.1 Hz). (D) Tone of the palmaris longus muscle have a negative weak correlation (r = −0.31) with body mass index. (E) Stiffness of flexor digitorum superficialis exhibit a significant decrease from normal weight to obesity category. (F) Stiffness of flexor digitorum superficialis have a negative weak correlation (r = −0.18) with body mass index. (G) Tone of the flexor digitorum superficialis did not show differences by weight status. (H) Tone of the flexor digitorum superficialis have a negative weak correlation (r = −0.15) with body mass index. *$p < 0.5$, **$p < 0.01$.

correlation between the stiffness of the flexor digitorum superficialis muscle (r = 0.18, $p < 0.001$) and BMI (Fig. 3F). We did not find significant differences in the tone of the flexor digitorum superficialis muscle by weight status ($F_{(132,2)} = 1.9$, $p > 0.05$) (Fig. 3G), although a weak negative correlation was found between the tone of the flexor digitorum superficialis muscle (r = 0.15, $p < 0.01$) and BMI (Fig. 3H).

## Mechanical properties of Achilles tendon, weight status, and BMI

Figure 4 illustrates the relationship between the mechanical properties of the forearm muscles and Achilles tendon, considering weight status and the influence of BMI. A one-way ANOVA showed a significant increase in the stiffness of the Achilles tendon from the normal-weight group to the obesity group (normal weight: 728.3 ± 72.6 N/m; obesity: 832.7 ± 124.1 N/m) ($F_{(132,2)} = 4.89$, $p < 0.001$) (Fig. 4A). Conversely, a one-way ANOVA did not show significant differences in the tone of the Achilles tendon between the normal-weight group and other weight statuses ($F_{(132,2)} = 1.58$, $p > 0.05$) (Fig. 4B). There was a significant weak positive correlation between the stiffness of the Achilles tendon (r = 0.34, $p = 0.0001$) and BMI (Fig. 4C), while a weak positive correlation was found between the tone of the Achilles tendon (r = 0.18) and BMI (Fig. 4D).

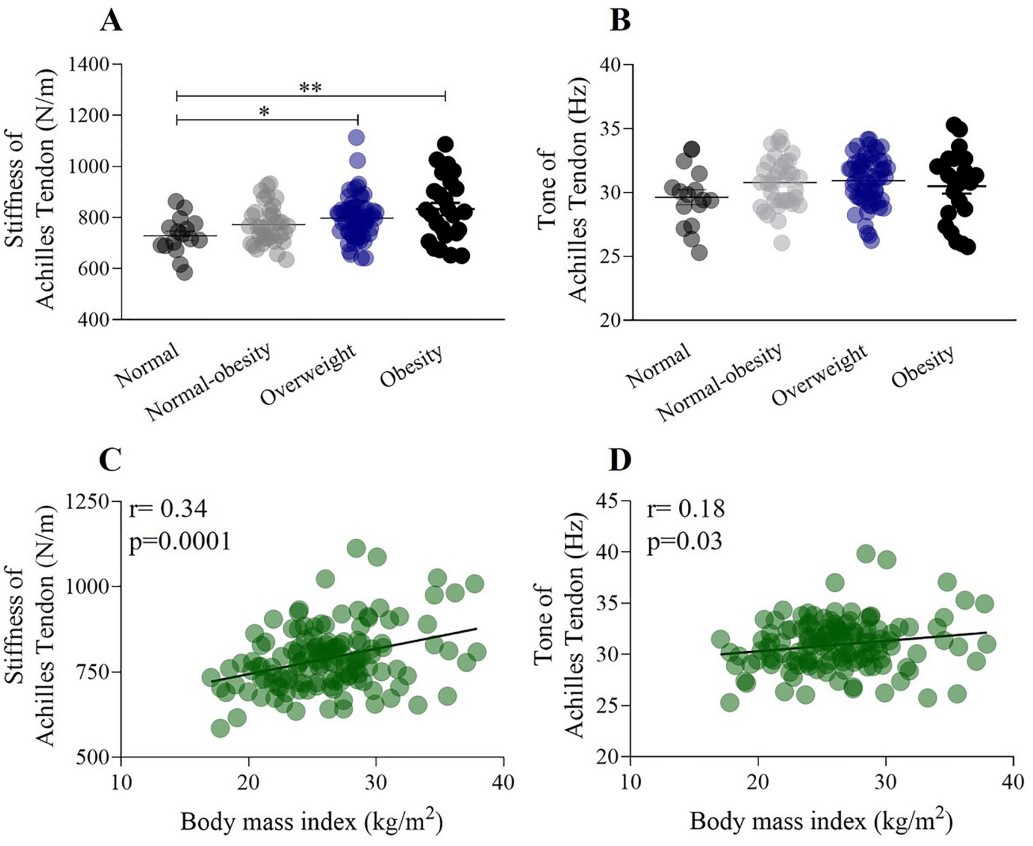

**Figure 4 The relationship between stiffness of the Achilles tendon, weight status, and the influence of BMI.** (A) Stiffness of the Achilles tendon significantly increased from the normal weight to the obesity category (normal weight: 728.3 ± 72.6 N/m; obesity: 832.7 ± 124.1 N/m). (B) The tone of the Achilles tendon did not change according to weight status. (C) Stiffness of the Achilles tendon shows a significant positive correlation with BMI (r = 0.34). (D) The tone of Achilles tendon shows a very weak correlation with BMI (r = 0.18). *$p < 0.5$, **$p < 0.01$.

## Multiple correlations among BMI, body composition, and the mechanical properties of the forearm muscle and Achilles tendon

Table 2 presents a multiple regression analysis examining the effect of BMI and body composition on the stiffness and tone of the forearm muscles and Achilles tendon. The relationship between stiffness, body composition, and BMI is weaker in the Achilles tendon compared to the forearm muscles (multiple correlation coefficient (R): forearm muscles, 0.34 ± 0.02; Achilles tendon, 0.25 ± 0.01). The regression model has a 34% predictive accuracy for the stiffness of the forearm muscles based on body composition and BMI, and nearly 25% predictive accuracy for the stiffness of the Achilles tendon. The tone of the palmaris longus shows differences depending on fat or lean mass content (multiple correlation coefficient (R): fat mass, 0.30 ± 0.03; lean mass, 0.19 ± 0.02). There was no significant correlation between BMI, body composition, and the tone of the Achilles tendon.

**Table 2  Multiple correlation analysis between BMI, body composition, and muscle or tendon mechanical properties.**

| Parameters | Body composition * BMI | Palmaris longus | | | Flexor superficialis | | | Achilles tendon | | |
|---|---|---|---|---|---|---|---|---|---|---|
| | | R | $R^2$ | p | R | $R^2$ | p | R | $R^2$ | p |
| Stiffness | Body fat mass | 0.369 | 0.13 | 0.0001 | 0.366 | 0.13 | 0.0001 | 0.244 | 0.05 | 0.0009 |
| | Percentage body fat | 0.337 | 0.14 | 0.0001 | 0.341 | 0.11 | 0.0002 | 0.264 | 0.07 | 0.0006 |
| | Skeletal muscle mass | 0.341 | 0.11 | 0.0002 | 0.321 | 0.10 | 0.0007 | 0.263 | 0.07 | 0.0001 |
| | Fat free mass | 0.338 | 0.11 | 0.0003 | 0.309 | 0.09 | 0.001 | 0.267 | 0.07 | 0.0001 |
| Tone | Body fat mass | 0.307 | 0.09 | 0.001 | 0.299 | 0.09 | 0.0019 | 0.09 | 0.008 | 0.57 |
| | Percentage body fat | 0.357 | 0.12 | 0.0001 | 0.279 | 0.07 | 0.0045 | 0.19 | 0.04 | 0.078 |
| | Skeletal muscle mass | 0.193 | 0.03 | 0.079 | 0.229 | 0.05 | 0.027 | 0.07 | 0.0005 | 0.708 |
| | Fat free mass | 0.190 | 0.03 | 0.08 | 0.218 | 0.04 | 0.039 | 0.078 | 0.0062 | 0.65 |

**Note:**
R, Multiple correlation coefficient; $R^2$, Coefficient of determination.

## DISCUSSION

The aim of this investigation was to determine the relationship between BMI and the mechanical properties of muscle and tendon, and to compare these properties across different weight statuses. First, we observed that relative HGS significantly decreases as weight status increases, showing a moderate negative correlation with BMI. Second, the stiffness and tone of the forearm muscles decreased with increasing weight status and BMI. Third, we found that the stiffness and tone of the Achilles tendon increased as weight status and BMI rose. In this study, the decision to focus on women, who generally have a higher body fat content than men, was deliberate, in order to analyze the effect of higher fat content on the mechanical properties of forearm muscle and tendon, and to compare its impact on strength performance.

Our study reveals that relative HGS decreased significantly from healthy weight to obese participants. Additionally, relative HGS showed moderate negative correlations with BMI across all participants; that is, as BMI increases, grip strength decreases. In a study comparing handgrip strength and BMI in female basketball players, a low correlation was found between BMI and handgrip strength. However, it should be noted that *Pizzigalli et al. (2017)* studied only normal-BMI females who also engage in sports, which differs from our sample. Our study sample consists exclusively of women, with 11.8% having normal weight, 25.7% with normal-weight obesity, 44.1% overweight, and 18.4% obesity. Furthermore, we found that relative HGS had a moderate negative correlation with body fat mass and a low correlation with fat-free mass (FFM), indicating that excess fat has a disadvantageous effect on handgrip strength, while the amount of FFM in our participants had no significant impact on grip strength. In this context, *Miller et al. (1993)* indicated that the differences in strength linked to sex are more pronounced in the upper body. Therefore, the differences in grip strength can be attributed to the fact that women generally have less lean body mass in the upper body (*Miller et al., 1993*). Interestingly, we found that relative HGS was significantly lower in the normal-obesity group compared to the normal-weight group. Excessive body fat, even in individuals with a healthy body

weight, is described as normal-weight obesity syndrome (*Oliveros et al., 2014*). A high body fat content, even within a normal body weight range, results in low fat-free mass, including skeletal muscle. In such cases, higher adiposity may reflect lower functional muscle mass (*Lafortuna et al., 2004*), and the risk of cardiometabolic dysregulation and systemic inflammation has been reported. Indeed, evidence indicates that fat infiltration into skeletal muscles is associated with lower muscle strength, reduced muscle power, and impaired physical function (*Hilton et al., 2008*). These findings suggest that muscle weakness could be caused by muscular fat infiltration.

We found significant differences in the percentage of body fat mass between the normal weight group and all other categories. Healthy muscle contains about 1.5% intramyocellular fat, which can increase to over 5% in individuals with obesity (*Malenfant et al., 2001*). Therefore, the inclusion of intramuscular fat is an important factor that affects muscle contractile performance, leading to a decrease in muscle force (*Rahemi, Nigam & Wakeling, 2015*) and potentially causing changes in the viscoelastic properties of the muscle (*Usgu, Ramazanoğlu & Yakut, 2021*).

Our study reveals a correlation between BMI and the stiffness and tone of the palmaris longus. Additionally, we found a weak correlation between BMI and the stiffness and tone of the flexor digitorum superficialis. Research on the relationship between BMI and muscle stiffness is limited and controversial. For instance, recent studies reported no association between BMI and muscle stiffness at rest or during contraction (*Hoffman et al., 2021*). However, it should be noted that *Hoffman et al. (2021)* study involved participants with a mean BMI of 23.51, corresponding only to individuals with a healthy weight status, and included an older age range (18 to 50 years) compared to our sample. In another study, the relationship between BMI and muscle stiffness in neck muscles showed a correlation in only one muscle group among multiple neck muscles measured. However, *Kuo et al. (2013)* studied only individuals classified as normal weight or underweight. *Usgu, Ramazanoğlu & Yakut (2021)* found weak positive correlations between BMI and the tone and stiffness of the bilateral biceps femoris, as well as the stiffness and elasticity of the right biceps brachii, across all participants. Notably, their sample included individuals ranging from normal weight to overweight categories. *Kocur et al. (2017)* explored the relationship between BMI and the stiffness and elasticity of the sternocleidomastoid muscle in females, reporting a strong correlation with elasticity and a moderate correlation with stiffness. Although this study included participants across all BMI categories (18.5 to 33.8) and only healthy women, the age range was very broad, spanning from 21 to 88 years. It is well-documented that body composition changes with age, even in the absence of changes in body weight. Aging leads to a loss of muscle mass, an increase in body fat, and fat infiltration into muscles (*Kim & Won, 2022*; *St-Onge & Gallagher, 2010*). In our study, we controlled for the age factor by using a limited age range (18–25 years) to compare biomechanical properties according to weight status and BMI. The mechanism affecting the stiffness and tone of superficial forearm muscles in relation to BMI is not entirely clear. Myotonometric measurements of muscle biomechanical properties depend on many variables related to the structure and function of connective tissue (*Kocur et al., 2017*). The
structure and morphology of myofascial tissue constantly and dynamically change in response to external loads (*Mayers, 2009*). It is known that extracellular matrix morphology, including the content and cross-linking of collagen (*Gosselin et al., 1998*), can influence stiffness. Therefore, we hypothesize that the most probable reason for increased stiffness is the infiltration of muscle connective tissue. Various studies have shown that the amount of non-contractile connective or adipose tissue in skin and myofascial tissues increases with fat content (*Liu et al., 2016*; *Sun et al., 2013*), which contributes to higher passive muscle stiffness and affects the quality of muscle contraction. Fat has stiffer material properties than muscle (*Rahemi, Nigam & Wakeling, 2015*); thus, the introduction of fat into muscle results in increased stiffness. Our results showed that the stiffness and tone of the palmaris longus and flexor digitorum superficialis muscles decreased with an increase in weight status, which partially contradicts (*Kelley, Goodpaster & Storlien, 2002*), who suggested that obese individuals have lower muscle quality, attributing the loss in contractile performance to increased muscle belly tissue stiffness as intramuscular fat concentration rises.

Apart from muscle properties, the properties and behavior of tendons are crucial for human movement and locomotion. Optimal tendon stiffness is essential for enhancing sports performance and preventing injuries (*Morgan et al., 2018*). In our study, Achilles tendon stiffness increased from normal weight status to obesity categories and was correlated with BMI. *Tomlinson et al. (2021)* explored the effects of obesity on skeletal muscle function and tendon properties, reporting that in young participants (18–49 years old) ranging from normal weight to obese, Achilles tendon stiffness positively correlated with both body mass and BMI. In another recent study, the relationship between Achilles tendon stiffness and BMI was investigated in individuals with a mean age of 20.19 years. No association was found between BMI and Achilles tendon stiffness (*Römer et al., 2023*). However, it should be noted that (*Römer et al., 2023*) studied only professional athletes, and the mean BMI was 22.85, which differs from our results.

We consider that the differences in Achilles tendon stiffness observed in this study may be attributed to fat mass content. We hypothesize that in young women, Achilles tendon stiffness adapts to the loading stimulus rather than responding directly to the nature of the load (*e.g.*, adiposity level). Our results are consistent with other studies reporting a higher correlation between Achilles tendon stiffness and increased BMI in young adults, though this was not observed in older individuals (*Tomlinson et al., 2021*). Moreover, it has been demonstrated that a higher BMI is associated with both a greater cross-sectional area of the Achilles tendon and increased stiffness in young individuals, with body mass strongly associated with both the cross-sectional area and stiffness of the Achilles tendon. This suggests that higher mechanical stress from excess body weight may lead to protective changes in tendon characteristics in obese individuals.

The present study has strengths and limitations. First, we chose to examine only a group of women. However, muscle strength and the content and distribution of adipose tissue differ between men and women (*Lafortuna et al., 2004*; *Ramírez-Vélez et al., 2018*), which may affect the interpretation of our results. Second, the examination was limited to the upper extremities for muscle assessments and only the Achilles tendon in the lower

extremities. Despite the large sample size, another limitation is the low number of participants in the normal-weight group (11.8%), while 88% had a body fat percentage over 30%. Although we found a relationship between the stiffness of forearm muscles and the Achilles tendon, the correlation ranged from weak to moderate. Additionally, the myometric method has limitations in investigating deep-seated but inaccessible muscles. Therefore, further research with different experimental groups is needed to corroborate our findings and gather additional data.

## CONCLUSIONS

The results of the present study support the notion that the stiffness of the forearm muscles and Achilles tendon is correlated with BMI in young adult women. Furthermore, higher grip strength is associated with increased muscle stiffness, while a higher body fat percentage is linked to decreased mechanical properties and poorer muscle function. Additionally, an increase in Achilles tendon stiffness is associated with a higher BMI.

### Funding
The authors received no funding for this work.

### Competing Interests
The authors declare that they have no competing interests.

### Author Contributions
- Miguel Ángel Pérez conceived and designed the experiments, performed the experiments, analyzed the data, authored or reviewed drafts of the article, writing and revising the entire article, and approved the final draft.
- Gabriela Urrejola-Contreras conceived and designed the experiments, performed the experiments, prepared figures and/or tables, authored or reviewed drafts of the article, and approved the final draft.
- Brian Alvarez performed the experiments, prepared figures and/or tables, and approved the final draft.
- Camila Steilen performed the experiments, prepared figures and/or tables, and approved the final draft.
- Antonieta Latorre performed the experiments, prepared figures and/or tables, and approved the final draft.
- Maximiliano A. Torres-Banduc conceived and designed the experiments, performed the experiments, analyzed the data, prepared figures and/or tables, authored or reviewed drafts of the article, and approved the final draft.

### Human Ethics
The following information was supplied relating to ethical approvals (*i.e.*, approving body and any reference numbers):

The University Viña del Mar granted Ethical approval to carry out the study within its facilities (Ethical Application Ref:: CEC-UVM 24-23).

## Data Availability

The data is available in the Supplemental File.

## Supplemental Information

Supplemental information for this article can be found online at http://dx.doi.org/10.7717/peerj.18430#supplemental-information.

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
