# Peer review of "Exploring the interplay between body mass index and passive muscle properties in relation to grip strength and jump performance in female university students"

_PeerJ, doi:10.7717/peerj.18430_

## Round 0.1 · original submission · Major Revisions

The manuscript seriously lacks a clear and compelling purpose, specifically in demonstrating how the study addresses a specific problem. The reported relationships appear self-evident and have been established in prior research. What unique insights or value does this paper provide that justify its contribution to the existing body of knowledge? Why is reading this paper worthwhile?

Reviewer 1 ·

Basic reporting

Main comments:

1. Introduction
- Please provide a clear research question and research perspective according to previous papers.
- I would suggest ordering different variables.
- Please describe the mechanical properties and benefits of using Mytonin in the evaluation of muscle/tendon properties.
- Please specify more clearly the study's aim and state hypothesis.

2. Discussion:
- Please shorten the first paragraph. Add a summary of the results and final findings from this study.
- Please refer to potential mechanisms for changes in mechanical properties and indicate an explanation due to the correlations. Finally, discuss it with some studies that found similarities and diverse observations.

3. Conclusions:
- Please re-write this section to indicate findings and potential explanations. Please state a final statement for future perspective and research.

Experimental design

Main comments:

1. Introduction
- Please provide a clear research question and research perspective according to previous papers.
- I would suggest ordering different variables.
- Please describe the mechanical properties and benefits of using Mytonin in the evaluation of muscle/tendon properties.
- Please specify more clearly the study's aim and state hypothesis.

2. Discussion:
- Please shorten the first paragraph. Add a summary of the results and final findings from this study.
- Please refer to potential mechanisms for changes in mechanical properties and indicate an explanation due to the correlations. Finally, discuss it with some studies that found similarities and diverse observations.

3. Conclusions:
- Please re-write this section to indicate findings and potential explanations. Please state a final statement for future perspective and research.

Validity of the findings

Main comments:

1. Introduction
- Please provide a clear research question and research perspective according to previous papers.
- I would suggest ordering different variables.
- Please describe the mechanical properties and benefits of using Mytonin in the evaluation of muscle/tendon properties.
- Please specify more clearly the study's aim and state hypothesis.

2. Discussion:
- Please shorten the first paragraph. Add a summary of the results and final findings from this study.
- Please refer to potential mechanisms for changes in mechanical properties and indicate an explanation due to the correlations. Finally, discuss it with some studies that found similarities and diverse observations.

3. Conclusions:
- Please re-write this section to indicate findings and potential explanations. Please state a final statement for future perspective and research

Additional comments

Main comments:

1. Introduction
- Please provide a clear research question and research perspective according to previous papers.
- I would suggest ordering different variables.
- Please describe the mechanical properties and benefits of using Mytonin in the evaluation of muscle/tendon properties.
- Please specify more clearly the study's aim and state hypothesis.

2. Discussion:
- Please shorten the first paragraph. Add a summary of the results and final findings from this study.
- Please refer to potential mechanisms for changes in mechanical properties and indicate an explanation due to the correlations. Finally, discuss it with some studies that found similarities and diverse observations.

3. Conclusions:
- Please re-write this section to indicate findings and potential explanations. Please state a final statement for future perspective and research.

·

Basic reporting

Hand grip test and CMJ tests are very common tests. How is the study here different from previous studies?. The study aimed to identify "mechanical properties of the forearm and Achilles tendon". In addition, nutritional status was investigated. How can there be a relationship between nutritional status and mechanical properties of the forearm and Achilles tendon? If nutritional status is to be investigated, it should be investigated as a pre-test and post-test. The status determination here needs to be scientifically explanatory.
The originality of the study should be written. It should be explained better. What makes this study different from others?
Is not "increased body fat percentage correlates with diminished mechanical properties" already expected, known and predicted?

Experimental design

The study was between 18 and 32 years (mean 19.8±2.3). While there is a range between 18 and 32 years, the average age is 19.8. Outliers related to age can be removed (such as the last 20% of the age group, 29 and above). There is a lot of range, but there is a clustering on one side. The data should be checked.
Myotonometric assessment of muscle mechanical properties and procedure: Adequate.
Statistical analysis
Homogeneous için Spearman yerine Levene yapılmalı. Homoscedasticity mi yoksa Homogeneous? Attention should be paid to spelling and scientific usage. Line 194: independent t test should be written instead of unpaired test.
Statistical method ANOVA was used. In writing, one way ANOVA is written. ANOVA means Analysis of variance. The correct spelling should be Analysis of variance (ANOVA) or just ANOVA.

Validity of the findings

"For CMJ force and power, there was a significant increase from the normal weight group to the obesity group. However, when expressed as a relative value, CMJ force and power show a significant decrease from the normal weight group to the obesity group". Aren't the findings in these two sentences the same? Why are the two sentences connected, however?
Why was BMI not used instead of nutritional status? Nutritional status should be explained. However, BMI was given in the study.
Table 1. Demographics, body composition, grip strength, and physical activity level. How was PA level measured? Until now, there has been no measurement or definition of PA. But it is in the table as PA level. The table title should be changed.
Percent body fat (%): normal: 24.3; normal obesity: 33.7
Fat free mass: N: 39.3; NO: 38.3.
SMM kg: N: 21.4; NO: 20.9. According to these values, a deviation is observed. What do the authors attribute this to? Could it be related to height. Height N: 162; NO: 158.9. Could this be related to this condition? There seems to be a discrepancy. This should be explained.
There is a difference between the values. But it is not clear in the tables in whose favor. Bonferoni results should be shown by marking a-b on the averages.
Table 2-3. Correlation value r and p should be given in the bottom rows.

Additional comments

Line 280-291. Conclusions are given in the discussion section. The results should be included in the conclusion section.
Line 285. If there is a negative correlation, which value increased and which decreased. It should be written.
Line 292. Muscle strength or hand grip strength?
There are deficiencies in the discussion section. The reason for the findings should be stated. What does the author relate the findings to?

Reviewer 3 ·

Basic reporting

The authors should review and correct grammar, as well as ensure consistent terminology throughout every part of the manuscript.

For example, there is a lack of abbreviations in several places, such as HGS.

On line 27, "impact" could be changed to "effect."
On line 50, add "the" intrinsic material properties.
On line 61, "has not effect" should be "has no effect."
On line 68, "their" should be "its,"

Especially in the abstract, abbreviations should be used. Additionally, detailed reporting of statistical analysis methods and statistical values in the results section should be included to ensure consistency with the report writing.

In the introduction section, the author lacks citations in several parts, such as at the end of the sentences on lines 52, 79, and 85. Additionally, in many parts where multiple previous works are referenced, only a single source is cited, such as on lines 65 and 91. The authors should include additional references.

In terms of writing structure, the authors should ensure consistency between the abstract and the introduction. For instance, in the abstract, HGS is mentioned before stiffness. It might be better to first discuss HGS and jump performance, and then mention stiffness tone and body composition. This would make the text more concise and connect all the gaps within a single paragraph before moving on to the objectives.

In the final paragraph, how does the researchers formulate the research hypothesis and the expected benefits from this study? Please specify this part further.

Experimental design

The authors should add information on the study design, specifying the experimental design methods used.

Why did the researchers only collect data from female participants? This should be justified, possibly within the introduction. Additionally, inclusion and exclusion criteria for the sample group should be reported, along with current background information about the participants.

In this study, were there sessions that allowed the participants to familiarize themselves with the experimental procedures and equipment? If so, please specify, and indicate the intervals between each session.

The researchers lack citations for the testing methods used in several steps, such as HGS and Jump performance, as well as for the reliability values of previous tests or protocols. For example:

- What was the power calculation based on? Why were the relative values for both force and power expressed as (W/kg/m²)? Please explain.

- For HGS, which arm was used, or were both arms tested? Was there a reference for each participant indicating whether tone and stiffness were measured on the same arm? Please specify.

- How was the nutritional status determined?

- Did the researchers assess the reliability of the measurements in this study, particularly for tone and stiffness, which are key variables? If so, please specify.

Validity of the findings

Did the researchers believe that having an unequal number of n in each category affects the discussion and conclusion of the experiment for other variables? This should be further explained in the discussion section. Additionally, is a sample group with an age range up to 32 years still considered 'Young'? Please explain.

Is the gender of the sample group considered a limitation of this study? Please explain.

The discussion should clearly reflect your findings, relate them to the literature, and explain their physiological or practical significance. Each paragraph should follow a logical sequence, as the current order is a bit hard to follow. The order of the discussion section may be adjusted according to the revisions of the introduction that I have previously suggested.

Additional comments

References
- Please make sure your references are in proper format and check PeerJ literature base for related papers for connection for this line of research in the journal.

---

## Round 0.2 · accepted · Accept

In my opinion, the reviewer comments were addressed satisfactorily. Reviewer 3 still issued some recommendations, but in my opinion they are optional.

Reviewer 1 ·

Basic reporting

I accept the revision.

Experimental design

I accept the revision.

Validity of the findings

I accept the revision.

Additional comments

I accept the revision.

·

Basic reporting

No comment.

Experimental design

No comment.

Validity of the findings

No comment.

Additional comments

No comment.

Reviewer 3 ·

Basic reporting

In all sections of the article, the author should check the use of abbreviations. If abbreviations have been introduced, they should replace the full terms consistently throughout. For example, both the full terms and abbreviations of HGS and CMJ are used repeatedly in the statistical analysis section.

Experimental design

The author should include the reliability data for each test in the respective sections. For example, the values for MyotonPRO, as mentioned in the response to the reviewer, can be added. For other equipment, reliability values from previous research, such as for the InBody270, can also be used.

Validity of the findings

In the abstract section, the author should provide more details on the statistical analysis, such as the use of multiple regression analysis and the significance levels set. This should be linked to the results section, where p-values and r-values should be reported to demonstrate the statistical significance.